# Intestinal Microbiota and Metabolomics Reveal the Role of *Auricularia delicate* in Regulating Colitis-Associated Colorectal Cancer

**DOI:** 10.3390/nu15235011

**Published:** 2023-12-04

**Authors:** Lanzhou Li, Honghan Liu, Jinqi Yu, Zhen Sun, Ming Jiang, Han Yu, Chunyue Wang

**Affiliations:** 1Engineering Research Center of Chinese Ministry of Education for Edible and Medicinal Fungi, Jilin Agricultural University, Changchun 130118, China; lilanzhou@jlau.edu.cn (L.L.); yujinqi@mails.jlau.edu.cn (J.Y.); sunzhen23@mails.jlau.edu.cn (Z.S.); 2School of Life Sciences, Jilin University, Changchun 130012, China; honghan20@mails.jlu.edu.cn; 3College of Life Science and Technology, Mudanjiang Normal University, Mudanjiang 157011, China; jiangming@mdjnu.edu.cn; 4College of Agriculture, Jilin Agricultural University, Changchun 130118, China

**Keywords:** colitis-associated colorectal cancer, *Auricularia delicate*, intestinal microbiota, NF-kB, inflammation

## Abstract

Background: The edible fungus *Auricularia delicate* (ADe) is commonly employed in traditional medicine for intestinal disorders; however, its inhibitory effect on colitis-associated colorectal cancer (CAC) and the underlying mechanisms remain unexplored. (2) Methods: The inhibitory effect of ADe on CAC was investigated using a mouse model induced by azoxymethane/dextran sulfate sodium. Results: ADe effectively suppressed the growth and number of intestinal tumors in mice. Intestinal microbiota analyses revealed that ADe treatment increased *Akkermansia* and *Parabacteroides* while it decreased *Clostridium*, *Turicibacter*, *Oscillospira*, and *Desulfovibrio*. ADe regulated the levels of 2′-deoxyridine, creatinine, 1-palmitoyl lysophosphatidylcholine, and choline in serum. Furthermore, the levels of these metabolites were associated with the abundance of *Oscillospira* and *Paraacteroides*. ADe up-regulated the free fatty acid receptor 2 and β-Arrestin 2, inhibited the nuclear factor kappa B (NF-κB) pathway, and significantly attenuated the levels of inflammatory cytokines, thereby mitigating the inflammatory in CAC mice. Conclusions: The protective effect of ADe in CAC mice is associated with the regulation of intestinal microbiota, which leads to the inhibition of NF-kB pathway and regulation of inflammation.

## 1. Introduction

The term colitis-associated colorectal cancer (CAC) refers to a subtype of colorectal cancer that develops due to long-term inflammation in the gastrointestinal tract, specifically associated with inflammatory bowel disease (IBD), encompassing Crohn’s disease (CD), and ulcerative colitis (UC) [1]. The prevalence of colorectal cancer (CRC) in UC patients accumulates to 7% after 30 years of disease [2]. Patients with CAC typically undergo standard surgical treatment; however, surgery may potentially result in intestinal damage, thereby impacting the area of intestinal absorption [3,4]. The mitigation of CAC can be achieved through pharmacotherapy aimed at reducing inflammation, such as the administration of oxaliplatin and fluorouracil during chemotherapy [5,6], which is responsible for diarrhea, nausea, absorption, and digestive disorders [7,8].

Accordingly, nuclear factor kappa B (NF-κB) activation due to chronic inflammation leads to overabundant tissue regeneration, thus promoting the production and development of tumor cells [9,10]. Tumor necrosis factor α (TNF-α) is a key factor in the development and advancement of IBD. Its inhibition through anti-TNF-α therapy has become a common approach for managing this condition [11]. The specific binding between TNF and its receptor can activate NF-κB, thus facilitating the development of CAC [12]. In the process of inflammatory reaction, reactive oxygen species (ROS) generated by inflammation induce DNA damage and mutations through oxidative reactions [10,13].

In clinical, intestinal microorganisms were considered to directly inflict epithelial injury and trigger “intestinal permeability”, thereby instigating chronic inflammation and further exacerbating tumor progression [14]. The mechanism of microorganisms-induced colon cancer mainly includes three hypotheses, including the α-bug hypothesis (cancer-promoting microorganisms that can produce toxins directly cause epithelial damage and promote the development of CAC) [15], common ground hypothesis (under the influence of both internal and external factors, such as diet and pollution, intestinal tissue damage increased the probability of microbial infection, thereby promoting the development of chronic inflammation and cancer in susceptible individuals) [16], and driver–passenger hypothesis (based on the α-bug hypothesis, the opportunistic pathogens and the induction of tumorigenesis are enhanced) [17]. As a biomarker of therapeutic efficacy, a hindrance to cancer treatment, and a probiotic for maintaining intestinal function, intestinal microbiota has received increasing attention in tumor clinical treatment [18]. The involvement of certain intestinal microorganisms in the development of enteritis and colon cancer has gradually come to light; for instance, genus *Desulfovibrio* has been found to compromise the integrity of the epithelial barrier while promoting cellular mutation [19,20]. The genus *Parabacteroides* is a significant source of short-chain fatty acids (SCFAs) in the intestine, which effectively inhibit inflammatory reactions and contribute to maintaining host-intestinal homeostasis [21]. SCFAs are considered active metabolites produced by intestinal microorganisms that regulate various physiological responses in the body, including anti-inflammatory and anti-tumor effects. By promoting SCFAs secretion and subsequently inhibiting NF-κB pathway activity, it aids in alleviating mouse colitis, reducing inflammatory cytokine expression, and facilitating mucosal repair [22]. The administration of chitooligosaccharides has been reported to protect mice from CRC by rectifying bacterial imbalance, specifically through the reduction of *Turicibacter* genus [23]. In mice, glycomacopeptide can elevate SCFAs levels and alleviate inflammation by decreasing *Desulfovibrio* genus [24]. Although regulating the intestinal microbiota is considered a promising research direction for cancer treatment, the role of intestinal microbiota in tumor clinical treatment is not fully common and collectively recognized. Modulating intestinal microbiota is still not a standard treatment option in clinical practice for cancer.

Based on its abundant biological activity and proven safety, mushrooms have the potential to emerge as a novel therapeutic option for addressing CAC. *Auricularia delicate* (ADe), commonly known as Deer tripe mushroom in China, belongs to the family Auriculariaceae. In both China and India, ADe is utilized as a functional food for managing gastrointestinal disorders [25,26]. The extract derived from ADe demonstrates potent antibacterial and antioxidant properties [25]. In vitro studies have shown that ADe polysaccharide effectively scavenges ROS, highlighting its antioxidative effects [27]. Our previous investigation revealed that ADe can exert anti-obesity effects in mice fed with high-fat diets by modulating oxidative stress [28]. However, the inhibitory effect of ADe on CAC, along with a lack of comprehensive understanding of underlying mechanisms.

This study utilized a murine model of azomethane (AOM) and dextran sodium sulfate (DSS)-induced CAC to assess inhibitory effects and underlying mechanisms of ADe on CAC development in mice. ADe effectively attenuated inflammatory responses and suppressed tumor growth in CAC mice by regulating intestinal microbiota composition and regulating the NF-κB signaling pathway. These findings contribute to the potential utilization of ADe as a possible therapeutic method for anti-CAC purposes.

## 2. Materials and Methods

### 2.1. Animal Experimental Protocol

This protocol adheres to the guidelines set by ARREAT and has been approved by the Animal Ethics Committee of Jilin University (SY202104008). The fruiting bodies of ADe were cultivated and collected at Jilin Agricultural University.

Forty male C57BL/6 mice, aged 7 weeks, were procured from Liaoning Changsheng Biotechnology Co., Ltd. [SCXK(Liao)2020-0001] (Benxi, China). The mice were kept in SPF environment with a controlled temperature (23 ± 1 °C), humidity (50 ± 10%), and a light/dark cycle of 12/12 h (light: 7:00–19:00). They were provided with unrestricted access to food and water. After one week of acclimatization, thirty mice were randomly selected for intraperitoneal injection with AOM (10 mg/kg physiological saline solution) (#A5486, Sigma-Aldrich, MO, USA) on the first day. The entire animal protocol lasted 10 weeks, during which drinking water was replaced with DSS (2% dissolved in water) (#S14049, Shanghaiyuanye Bio-Technology Co., Ltd., Shanghai, China) in weeks 2, 5, and 8. Starting from the fifth week onwards, the aforementioned thirty mice were separated into three groups (*n* = 10): a model group receiving oral saline and two ADe-treated groups receiving daily doses of either 0.5 (ADe_L_) or 1.0 g/kg ADe (ADe_H_), respectively, for six weeks. The remaining ten mice served as the control group, receiving intraperitoneal injections of normal saline followed by regular drinking water.

From the fifth week onwards, control mice received daily oral saline treatment for six weeks (Figure 1A). Blood samples were obtained from the tail veins of all mice at the last administration two hours later. They stood at 25 °C for 30 min before being centrifuged twice at 3000 r/min for 10 min to prepare serum. After euthanizing the mice through carbon dioxide inhalation, the cecal contents were collected under sterile conditions, while colorectum, liver, kidney, spleen, and heart tissues were immediately collected. The colorectum index is the ratio of the weight to length of the colorectum, which is used to compare the proliferation of intestinal tissue. Organ index (including liver, spleen, kidney, and heart) is the ratio of organ weight to body weight. Some of these tissues were fixed in 4% paraformaldehyde for subsequent histopathological examination. The remaining tissue samples, along with serum and cecal contents, were stored at −80 °C.

### 2.2. Histopathological Examination

Following the methodology described in a previous study [29], after fixation in 4% paraformaldehyde for 48 h, colorectal, liver, spleen, kidney, and heart tissues from mice (*n* = 3 per group) were sequentially treated with an ethanol solution, xylene, and paraffin to obtain paraffin-embedded tissues. Subsequently, the tissues embedded in paraffin were sliced into 5 μm sections. These sections underwent dewaxing with xylene and ethanol before being stained with hematoxylin and eosin (H&E) and dehydrated through successive treatments of ethanol and xylene. Finally, the sections were observed under a microscope (ECLIPSE E100, Nikon, Tokyo, Japan).

### 2.3. Intestinal Microbiota Analysis

The cereal contents of four mice from each group were randomly selected and utilized for the analysis of intestinal microbiota. Nucleic acids were isolated from the contents of each cecum and underwent PCR amplification, specifically targeting the 16S rRNA V3-V4 region [30]. The resulting products were sequenced using a double-ended approach with 2 × 250 bp reads. Following the determination by Shanghai Personal Biotechnology Co., Ltd., the bacterial sequences were uploaded to NCBI SRA under the registered number PRJNA860221, which can be accessed via the provided website link (https://www.ncbi.nlm.nih.gov/sra/PRJNA860221, accessed on 23 July 2022). The sequencing data were clustered using DADA2 with a 100% similarity threshold. Based on the amplicon sequence variants (ASVs) data, a flower plot was created to compare the differences in ASVs composition between groups. The alpha diversity indices were utilized to assess variations in the uniformity, richness, and diversity of intestinal microbiota across different groups. The unweighted UniFrac distance matrix was calculated to compare the differences in intestinal microbiota distribution among groups based solely on the presence or absence of changes without considering abundance. The differences in dominant genus-level microorganisms and the abundance of dominant biomarkers in each group were compared using a speciation heat map and LEfSe analysis of the microorganism abundance. The prediction of microbial function can be achieved by employing phylogenetic surveys of communities through the reconstruction of unobserved states, as demonstrated in previous studies [31].

### 2.4. Metabolomics Analysis

The serum sample (100 μL) was added to a pre-cooled solution of methanol and acetonitrile (400 μL, 1:1, *v*/*v*), and vigorously shaken to ensure thorough mixing. The resulting mixture was then centrifuged, the supernatant removed, and the remaining solid dried. Prior to analysis using mass spectrometry, the samples were reconstituted in a solution containing equal parts of acetonitrile and water (1:1, *v*/*v*). The solution was subsequently separated using the Agilent Technologies 1290 Infinity UHPLC system (Santa Clara, CA, USA), equipped with an ACQUITY UPLC BEH Amide column (1.7 μm, 2.1 × 100 mm), and analyzed utilizing the AB Sciex TripleTOF 6600 quadrupole-time of flight mass spectrometer (Framingham, MA, USA). The separation conditions employed were consistent with previous literature [32]. After data processing, the abundance of signature metabolites was filtered using orthogonal partial least squares discriminant analysis with VIP > 1 and *p*-value < 0.05. Subsequently, a Venn diagram and heatmap were generated to visualize the results. Additionally, a combined analysis of microorganisms at the genus level (top 20 abundance) and significantly changed metabolites were used to plot an associated heatmap.

### 2.5. The Analysis of Inflammatory Cytokines in Tumor Tissue

The obtained colorectal tumor samples were partially segmented and mixed with phosphate buffer solution before being homogenized. Protein concentrations were determined using the BCA Kit (#23227, Thermo Fisher, Waltham, MA, USA). The expressions of interleukin (IL)−1α, IL-1β, IL-6, IL-17A, IL-27, interferon (IFN)-γ, IFN-β, TNF-α, and monocyte chemoattractant protein-1 (MCP-1) in tumors were measured using the LEGENDplex™ Mouse Inflammation Panel (#740446 Biolegend, San Diego, CA, USA). The levels of IL-12 (#MM-0174M1), IL22 (#MM0892M1), and granulocyte macrophage colony-stimulating factor (GM-CSF) (#MM0185M1) in tumors were detected using standard assay kits from Jiangsu MEIMIAN (Yancheng, China).

### 2.6. Western Blotting

Tumor tissues were placed in RIPA Lysis Buffer (#20-188, Merck Millipore, Burlington, MA, USA) supplemented with a 1% Cocktail (#P002, NCM Biotechnology Co., Ltd., Suzhou, China) for protein extraction at 4 °C. The target proteins were separated using a 10% sodium dodecyl sulfate-polyacrylamide gel electrophoresis and subsequently transferred onto PVDF membranes. The membrane was blocked with a fast blocking solution (#GF1815, Culham Science Centre, Oxfordshire, UK) at 25 °C for 15 min. Primary antibodies (Appendix A) were incubated on the membrane at 4 °C for 12 h. After removing the primary antibody, corresponding secondary antibodies (Appendix A) were incubated on the membrane at 4 °C for 4 h [33]. The protein was reacted with Electrochemiluminescence kits (#GK10006, GLPBIO, Montclair, CA, USA) and detected using an imaging system (Tanon 5200, Tanon Science and Technology Co., Ltd., Shanghai, China). The level of pixel density was quantified using ImageJ v1.8.0 (National Institutes of Health, Bethesda, MD, USA).

### 2.7. Statistical Analysis

The data are presented as mean ± standard error of the mean (S.E.M.). Differences in data between groups were assessed using a post-hoc multiple comparisons (Dunnett) test (BONC DSS Statistics 25). Statistical significance for the differences in data between groups was defined as a *p*-value less than 0.05.

## 3. Results

### 3.1. ADe Inhibited the Tumor in CAC Mice

AOM/DSS treatment led to an extensive increase in the growth of colorectal tumors in mice, whereas ADe treatment significantly inhibited the incidence and progression of colon tumors without affecting body weight in CAC mice. This led to smaller tumor sizes and reduced number of tumors compared to untreated CAC mice (Figure 1B,D). The proliferation of intestinal tissue was evaluated using the colorectum index. AOM/DSS significantly increased intestinal tissue proliferation, while ADe restored colorectal proliferation without significant difference. ADe failed to change the structures of the liver, kidney, and heart in CAC mice (Appendix A). In CAC mice, ADe notably suppressed spleen index (*p* < 0.05) and megakaryocyte count in the spleen while maintaining its pathological structure (Appendix A). Colorectal tissues from CAC mice exhibited a high nucleocytoplasmic ratio, indistinct nucleoli, decreased infiltration of plasma cells, and increased necrotic cell fragments; these effects were all regressed by ADe administration (Figure 1D).

### 3.2. ADe Regulated Intestinal Microbiota in CAC Mice

The alterations in the intestinal microbiota directly influence the occurrence and progression of CAC [14]. Based on the Venn diagram, a total of 15,209 ASVs were detected, with 1595 belonging to all four groups. The unique ASVs in normal mice, CAC mice, ADe_L_-treated mice, and ADe_H_-treated mice were 4130, 3048, 3502, and 2934, respectively (Figure 2A). Principal coordinate analysis (PCoA) was employed to assess and compare the overall similarity of communities among groups; significant differences in microbial composition were noted between the CAC mice and the ADe treatment mice (Figure 2B). ADe_L_ treatment significantly increased alpha diversity, as indicated by the Simpson index (Figure 2C). The diversity in the top 20 abundances of intestinal microbiota between groups was analyzed using a genus-level heatmap. AOM/DSS treatment resulted in increased levels of *Clostridium*, *Turicibacter*, *Oscillospira*, and *Desulfovibrio*, which were subsequently suppressed by ADe treatment. The abundance of eight genera, particularly *Akkermansia* and *Parabacteroides*, was significantly increased in CAC mice following ADe treatment (Figure 2D and Appendix A). LEfSe analysis revealed that the model group exhibited three prominent taxa: *Clostridium celatum* at the species level, *HB2_32_21* at the genus level, and Alteromonadaceae at the family level. In contrast, the ADe-treated group showed six significant taxa: *Ruminococcus callidus* species, *Oscillospira* and *Paraprevotella* genera, Paraprevotellaceae and Burkholderiaceae family, as well as *Burkholderia* genus (Figure 2E). These findings confirm that ADe treatment has a notable influence on intestinal microbiota composition in CAC mice.

### 3.3. ADe Regulated Serum Metabolism in CAC Mice

The application of untargeted metabolomics enables a comprehensive and systematic analysis of metabolic profiles, facilitating the identification of differential metabolites. Notably, significant differences were observed in two specific metabolites across each group: 1-methylnicotinamide and cyclopiazonic acid (Figure 3A). In comparison to CAC mice, ADe exhibited a down-regulation of 16 serum metabolites, including 2′-deoxyuridine, allantoin and 1-palmitoyl lysophosphatidylcholine, and an up-regulation of 5 serum metabolites, including alpha-tocopherol (Vitamin E), 1-aminocyclopropanecarboxylic acid, nicotinamide N-oxide, 1-methylnicotinamide and cyclopiazonic acid (Figure 3B and Appendix A). The IBD marker, allantoin, exhibited a positive correlation with DL-indole-3-lactic acid and a negative correlation with 1-methylnicotinamide (*p* < 0.01) (Figure 3C). Correlation analysis was employed to examine the association between differential serum metabolites and differential intestinal microbiota. The genus *Oscillospira* exhibited a positive correlation with 2′-deoxyuridine (*p* < 0.05). Conversely, the genera *Lactobacillus* and *Allobaculum* displayed a negative correlation with 2′-deoxyuridine (*p* < 0.05). Furthermore, the genus *Paraprevotella* demonstrated a negative correlation with 1-palmitoyl lysophosphatidylcholine, nervonic acid, and behenic acid (*p* < 0.05). Similarly, the genus *Parabacteroides* exhibited a negative correlation with creatinine, 1-palmitoyl lysophosphatidylcholine, choline, nervonic acid, protocatechuic acid, and behenic acid (*p* < 0.05) (Figure 3D).

### 3.4. ADe Suppressed the Inflammation and NF-κB Signaling in CAC Mice

The significance of inflammation is crucial in the onset and advancement of CAC [34]. Compared to CAC mice, ADe-treated reduced IL-1α (>95%) (*p* < 0.001), IL-1β (by 81% at 1.0 g/kg ADe) (*p* < 0.001), IL-6 (by 88% at 1.0 g/kg ADe) (*p* < 0.001), IL-17A (>51%) (*p* < 0.001), TNF-α (>67%) (*p* < 0.001), and MCP-1 (by 64% at 1.0 g/kg ADe) (*p* < 0.01). Correspondingly, ADe significantly up-regulated the levels of IL-12 (by 93% at 1.0 g/kg ADe) (*p* < 0.05), IL-22 (by 141% at 0.5 g/kg ADe) (*p* < 0.001), IL-27 (by 563% at 0.5 g/kg ADe) (*p* < 0.001), IFN-γ (by 135% at 0.5 g/kg ADe) (*p* < 0.001), IFN-β (by 311% at 0.5 g/kg ADe) (*p* < 0.001), and GM-CSF (by 67% at 1.0 g/kg ADe) (*p* < 0.05) (Figure 4A–L). The regulatory effect of ADe on IL-1, IL-1, IL-6, IL-12, IL-17A, GM-CSF, MCP-1, and TNF is dose-dependent. However, the effect on IL-22, IL-27, IFN, and IFN in the 0.5 g/kg ADe treatment group is greater than that of 1.0 g/kg ADe. The above data demonstrate that ADe effectively attenuated inflammation in CAC mice.

Preclinical studies have found that some Parabacteroides species can reduce intestinal inflammation by producing SCFAs and inhibiting NF-kB signaling [21]. Based on the requirements of safety and accuracy of clinical treatment, it is necessary to clarify the exact mechanism of intestinal microorganisms for the application of corresponding therapeutic options in clinical. The binding ability of β-Arrestin 2 is regulated by metabolites produced by microorganisms through the activation of free fatty acid receptor 2 (FFAR2), resulting in a reduction in nuclear translocation of NF-κB [35,36]. The levels of FFAR2 were significantly increased by 0.5 g/kg ADe treatment (by 45%) (*p* < 0.05), along with β-Arrestin 2 (by 60%) (*p* < 0.05) and TAB1 (by 134%) (*p* < 0.05). The regulatory effect of 0.5 g/kg ADe on FFAR2, β-Arrestin 2, and TAB1 is greater than that of 1.0 g/kg ADe. Additionally, ADe treatment resulted in a reduction in the phosphorylated of inhibitor of nuclear factor kappa-B alpha (IκBα) (by 25% at 1.0 g/kg ADe) (*p* < 0.01), transforming growth factor-β activated kinase 1 (TAK1) (by 43% at 1.0 g/kg ADe) (*p* < 0.05), inhibitor of nuclear factor kappa-B kinase (IKK) (>36%) (*p* < 0.05), NF-κB (by 39% at 1.0 g/kg ADe) (*p* < 0.05), reduced the protein levels of IL-6 (>21%) (*p* < 0.05), IL-1β (>38%) (*p* < 0.01), IL-1α (by 24% at 1.0 g/kg ADe) (*p* < 0.05), TNF-α (>29%) (*p* < 0.05), and MCP-1 (by 44% at 1.0 g/kg ADe) (*p* < 0.01) in colorectal tissues (Figure 5). ADe regulated the expressions of associated proteins and suppressed the NF-κB signaling.

## 4. Discussion

This study confirmed that treatment with ADe reduced the occurrence and development of colorectal tumors and protected intestinal tissue cells in mice with CAC induced by AOM/DSS. Additionally, ADe treatment influenced the alterations in intestinal microbiota and serum metabolites of CAC mice while inhibiting inflammation mediated by the NF-κB pathway. According to reports, polysaccharide from *Pleurotus pulmonarius* inhibited carcinogenesis in CAC mice by regulating the NF-κB pathway and suppressing inflammation [37]. Similarly, the inhibitory effect of ADe on CAC mice may be achieved through adjusting intestinal microbiota and regulation of inflammation mediated by the NF-κB pathway.

Affected by lifestyle changes, the composition of the host’s intestinal microbiota undergoes significant alterations, subsequently impacting susceptibility to intestinal diseases [38]. Research on intestinal microbiota has gradually unveiled the correlation between microorganisms and CAC. The abundance of the genus *Turicibacter* is positively associated with inflammation levels in CAC mice [39]. As a sulfate-reducing bacterium, the genus *Desulfovibrio* can produce hydrogen sulfide, leading to dysfunction of the colorectal barrier and inflammation in both the colon and liver. This contributes to creating a microenvironment conducive to CRC tumor occurrence and metastasis [40]. Treatment with *Akkermansia* has been proven effective in reducing inflammation and improving LPS-induced dysfunction of the intestinal barrier by inhibiting activation of the NF-kB pathway [41,42]. *Akkermansia* selectively decreases in fecal microbiota among IBD patients, while supplementation with *Akkermansia* can regulate immune-killing effects and inhibit IBD as well as CAC development [43]. Both genera *Akkermansia* and *Parabacteroides* promote SCFAs production, which can be regulated through the NF-kB pathway to suppress inflammation and help maintain host intestinal homeostasis [21,44,45]. In AOM/DSS-induced CAC mice models, ADe treatment prevented an increase in *Clostridium*, *Turicibacter*, *Oscillospira,* and *Desulfovibrio* genera while also preventing a decrease in *Akkermansia* and *Parabacteroides* genera. These findings suggest that ADe treatment may protect intestinal tissue by improving the composition of intestinal microbiota.

The regulation of inflammation and host metabolism is significantly influenced by the intestinal microbiota. In this study, the combined analysis of intestinal microbiota and metabolomics revealed a positive correlation between the genus *Oscillospira* and 2′-deoxyuridine, as well as a negative correlation between the genus *Parabacteroides* and creatinine, 1-palmitoyl lysophosphatidylcholine, and choline. The detection of 2′-deoxyuridine can serve as an inflammatory biomarker to evaluate inflammatory damage [46,47]. An increase in creatinine levels reflects muscle damage and inflammation [48]. Moreover, 1-palmitoyl lysophosphatidylcholine acts as an effective inducer of inflammation by promoting white blood cell migration and increasing pro-inflammatory mediator levels [49]. The reduction of choline leads to a decrease in mitochondrial ATP synthesis, triggering AMPK activation, inhibiting NLRP3 activation, and reducing the production of IL-18 and IL-1β, thereby mitigating inflammation [50]. These conclusions align with the outcomes derived from analyzing intestinal microbiota, suggesting that in CAC mice, ADe may exert anti-inflammatory effects by synergistically regulating both intestinal microbiota and serum metabolites.

Chronic inflammation promotes cancer by facilitating the accumulation of ROS, regulating the tumor microenvironment, and activating NF-κB transcription [51]. In the context of human intestinal epithelial cells, the activation of NF-κB pathway by IL-1β leads to an upregulation in the expression of IL-6 [52]. The triggering of IL-1/IL-6 axis contributes to CAC development while blocking IL-1β activity demonstrates a substantial decrease in both mucosal damage and tumor development [53]. IL-1β can also promote colon cancer by inducing the IL-17 response [54], which leads to increased inflammation and oxidative stress levels that activate NF-κB [55], promoting the development of CAC [56]. The activation of NF-κB by TNF-α is believed to directly contribute to the development of CAC and the inhibition of TNF-α has been widely acknowledged as an effective therapeutic approach for patients with IBD [57]. MCP-1 expression is regulated by NF-κB, and elevated levels of MCP-1 pose an increased risk for IBD [58]. The IL-22 pathway partially mediates the process of anti-TNF-α treatment, while IL-22 treatment can attenuate inflammatory infiltration in colon tissue and enhance goblet cell abundance within the intestine [59]. The absence of GM-CSF can result in impaired innate immune response in the intestine, thereby increasing DSS-induced intestinal inflammation [60]. The high expression of IFN genes can impede the development of CAC by inducing pyroptosis in tumor cells [61]. The ADe treatment effectively suppressed the inflammatory levels in CAC mice, which is consistent with alterations observed in intestinal microbiota, metabolomics, and spleen histopathology. The above findings suggest that the protective effect of ADe in CAC mice may be attributed to the regulation of inflammation through the NF-κB pathway.

Based on previous research, SCFAs have been shown to activate FFAR2 and enhance the interaction between IκBα and β-Arrestin 2, thereby inhibiting NF-κB pathway. On the other hand, SCFAs increase the binding affinity of β-Arrestin 2 for TAB1, leading to further inhibition of TAB1-TAK1 binding, blockade of TAK1 phosphorylation, and subsequent IKK phosphorylation, ultimately resulting in reduced nuclear translocation of NF-κB [62,63]. The levels of these proteins in tumor tissues of CAC mice are influenced by ADe. Interestingly, the effect of ADe on some proteins is not dose-dependent, indicating that some active ingredients in ADe might have the best active dose. Based on our current data, the protective effect of ADe in CAC mice is associated with its regulation of intestinal microbiota, which subsequently inhibits NF-κB pathway activation and regulates inflammation levels.

The limitation of this study is that it only addresses a subset of intestinal microorganisms, and further systematic analysis is needed to understand the specific functions of individual microorganisms within the complex composition and functionality of the intestinal microbiota. The sample size of some results is limited. Increasing more samples to verify the results in further studies will be helpful in determining the activity mechanism of ADe.

## 5. Conclusions

The administration of ADe in CAC mice safeguards intestinal tissue by regulating the mouse intestinal and serum metabolites, inhibiting NF-kB pathway activation, and mitigating intestinal inflammation. These discoveries provide valuable perspectives on the possible therapeutic use of ADe in treating CAC.

## Figures and Tables

**Figure 1 nutrients-15-05011-f001:**
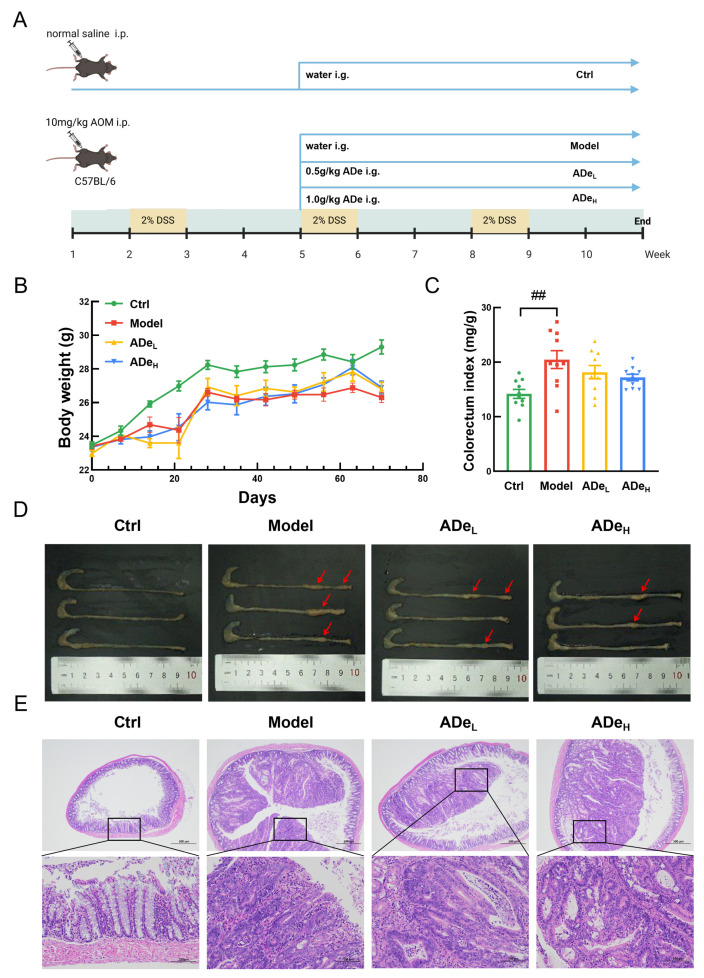
The protective effect of ADe in CAC mice. (**A**) A simplified flowchart depicting the animal experimental protocol. (**B**) Body weights of mice (*n* = 10). (**C**) Colorectum index measurements (*n* = 10). (**D**) Representative colorectal tissues of each group. (**E**) H&E pathological sections showing colorectal tumors at different magnifications (40× scale bar: 500 μm; 400× scale bar: 50 μm) (*n* = 3). ^##^
*p* < 0.01 vs. control group; red arrows: tumor tissue.

**Figure 2 nutrients-15-05011-f002:**
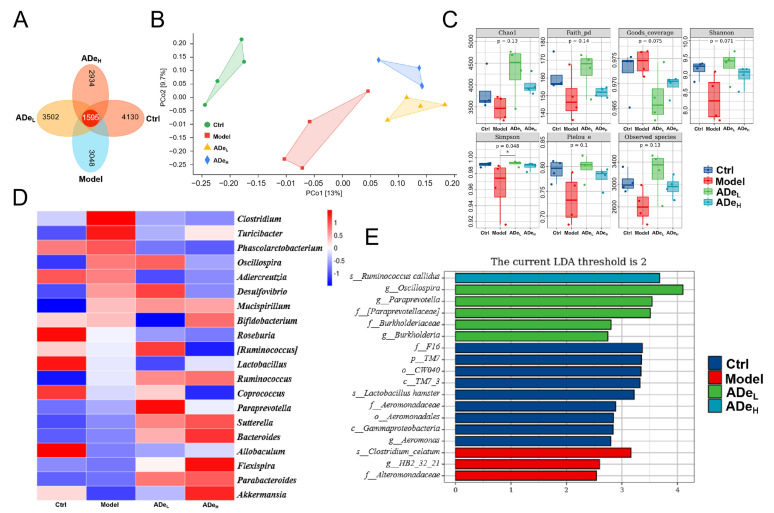
The intestinal microbiota of CAC mice (*n* = 4) is regulated by ADe treatment. (**A**) A Venn diagram illustrated the overlap between different microbial taxa. (**B**) PCoA of unweighted UniFrac distance was executed to assess beta diversity. (**C**) Grouping box plots were generated to compare alpha diversity indices. (**D**) A heatmap was constructed to display the composition of the top 20 dominant genera. (**E**) LEfSe analysis. The logarithmic score threshold for LDA analysis was set at 2.0.

**Figure 3 nutrients-15-05011-f003:**
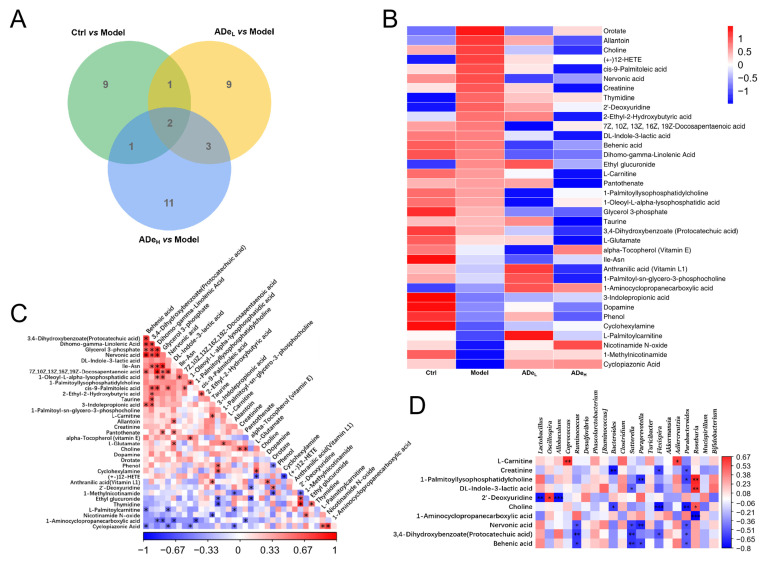
The levels of serum metabolites in CAC mice were regulated by ADe treatment. (**A**) Venn diagram illustrated the overlap of altered metabolites. (**B**) Heatmap displaying 36 significantly altered metabolites. (**C**) Heatmap showing the associated alterations in metabolite levels. (**D**) Heatmap presenting the associations between significantly altered metabolites and microbiota species. * *p* < 0.05, ** *p* < 0.01, *** *p* < 0.001.

**Figure 4 nutrients-15-05011-f004:**
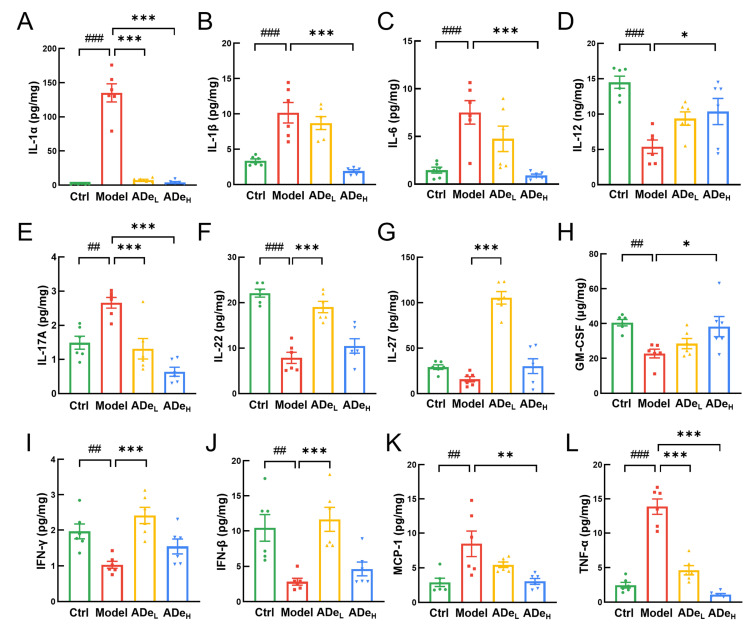
The regulatory effect of ADe on colorectal cytokines in colorectal tumor of CAC mice. Specifically, (**A**) IL-1α, (**B**) IL-1β, (**C**) IL-6, (**D**) IL-12, (**E**) IL-17A, (**F**) IL-22, (**G**) IL-27, (**H**) GM-CSF, (**I**) IFN-γ, (**J**) IFN-β, (**K**) MCP-1, and (**L**) TNF-α. (*n* = 6). ^##^
*p* < 0.01, and ^###^
*p* < 0.001 vs. control group; * *p* < 0.05, ** *p* < 0.01, and *** *p* < 0.001 vs. model group.

**Figure 5 nutrients-15-05011-f005:**
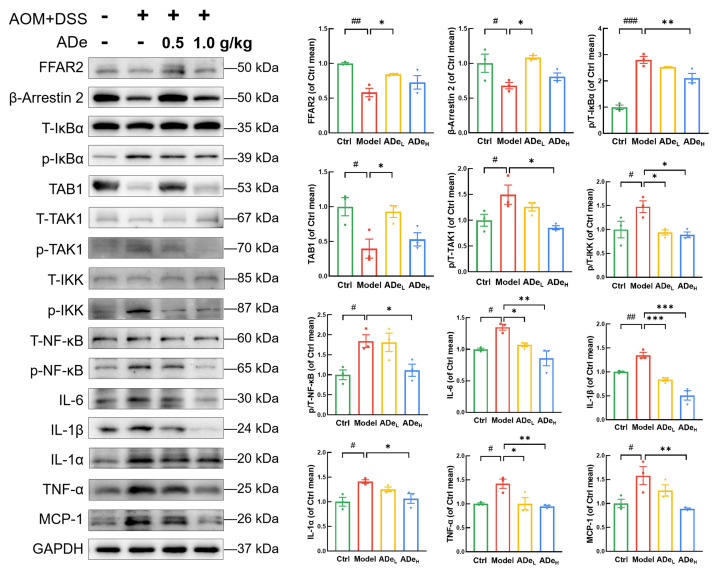
The proteins associated with the NF-κB pathway and inflammation in colorectal tumors of CAC mice. Quantification data were standardized by employing GAPDH as a reference, and fold change values were reported relative to control mice (*n* = 3). ^#^
*p* < 0.05, ^##^
*p* < 0.01, and ^###^
*p* < 0.001 vs. control group; * *p* < 0.05, ** *p* < 0.01, and *** *p* < 0.001 vs. model group.

## Data Availability

The bacterial sequences were uploaded to NCBI Sequence Read Archive under accession number PRJNA860221 (https://www.ncbi.nlm.nih.gov/sra/PRJNA860221).

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
