# Peer review of "Intestinal Microbiota and Metabolomics Reveal the Role of Auricularia delicate in Regulating Colitis-Associated Colorectal Cancer"

_nutrients, 2023, doi:10.3390/nu15235011_

Round 1

Reviewer 1 Report

Comments and Suggestions for Authors

In the review article by Li et al., entitled “Intestinal Microbiota and Metabolomics Reveal the Role of Auricularia delicate in regulating Colitis-associated Colorectal Cancer” reported the effects of the edible fungus Auricularia delicate on colorectal cancer. The authors used ADe, which can reduce intestinal inflammation by inhibiting the activation of the NF-kB pathway. This article is interesting to people working in the field of colon cancer. However, the authors need to address all the queries raised by the reviewers.

Comments:

Do the authors know the active component of Auricularia delicate that gives a protective effect against colon cancer?

The authors collect cecal contents from groups of four mice for the intestinal microbiota and metabolomics. Is this enough to derive the conclusion, given a small sample size?

How do Auricularia delicate modulating intestinal microbiota?

Does obesity increase the risk for colon cancer, and does Auricularia delicate have a protective effect against obesity-associated colon cancer? 

Comments on the Quality of English Language

The manuscript requires revisions of redundant sentences. Improving it will enhance its reliability.

Author Response

In the review article by Li et al., entitled “Intestinal Microbiota and Metabolomics Reveal the Role of Auricularia delicate in regulating Colitis-associated Colorectal Cancer” reported the effects of the edible fungus Auricularia delicate on colorectal cancer. The authors used ADe, which can reduce intestinal inflammation by inhibiting the activation of the NF-kB pathway. This article is interesting to people working in the field of colon cancer. However, the authors need to address all the queries raised by the reviewers.

Comments:

Do the authors know the active component of Auricularia delicate that gives a protective effect against colon cancer?

Reply: In this study, we investigated the anti-colorectal cancer activity and mechanism of Auricularia delicate, but the active component against colon cancer in Auricularia delicate is not clear. A large amount of research is needed to determine the active components against colon cancer in Auricularia delicate in future.

The authors collect cecal contents from groups of four mice for the intestinal microbiota and metabolomics. Is this enough to derive the conclusion, given a small sample size?

Reply: Sorry for the unclear description, the contents of the cecum were collected from four mice in each group. The corresponding content has been corrected. Intestinal microbiota analysis was used to screen for possible mechanisms, the sample size can meet the research needs, and ELISA and Western blotting were used to validate the possible mechanisms.

How do Auricularia delicate modulating intestinal microbiota?

Reply: Auricularia delicate treatment modulated the abundance of specific intestinal microbiota. Auricularia delicate-treated increased Akkermansia and Parabacteroides, while decreased Clostridium, Turicibacter, Oscillospira and Desulfovibrio. However, the specific mechanism by which Auricularia delicate regulates intestinal microbiota has not been studied in this study. More research is needed to prove the corresponding hypothesis.

Does obesity increase the risk for colon cancer, and does Auricularia delicate have a protective effect against obesity-associated colon cancer?

Reply: Obesity is a risk factor for colorectal cancer based on its molecular and metabolic effects on insulin, IGF-1, leptin, adipocytokines, sex hormones[1]. Our previous study has confirmed that Auricularia delicate treatment inhibited the obesity in mice[2], and this study has confirmed that Auricularia delicate treatment againsted colon cancer. Based on the above results, it can be inferred that Auricularia delicate may have a protective effect against obesity-associated colon cancer, but an obesity-associated colon cancer model is needed to establish in future study for validation to determine the activity and mechanism.

[1] I. Gribovskaja-Rupp, L. Kosinski, K.A. Ludwig, Obesity and colorectal cancer, Clinics in colon and rectal surgery 24 (2011) 229-243. https://doi.org/10.1055/s-0031-1295686.

[2] L.Z. Li, S.Y. Zhai, R.C. Wang, et al., Anti-Obesity Effect of Auricularia delicate Involves Intestinal-Microbiota-Mediated Oxidative Stress Regulation in High-Fat-Diet-Fed Mice, Nutrients 15 (2023) 15. https://doi.org/10.3390/nu15040872.

Reviewer 2 Report

Comments and Suggestions for Authors

Manuscript ID: Nutrients-2710795.

This study by Lanzhou Li and colleagues describes the effect of the fungus Auricularia delicate in DSS colitis cancer associated.

The strenght of the study are the results obtained (treatment supress inflammatory levels in CAC mice)

Nevertheless, there are some considerations the authors need to review.

MAJOR COMMENTS:

- Introduction:

- line 38-39: percentage regarding colorectal cancer in IBD is obtained old studies. Another review about this issue indicate a risk of about 7% at 30 years:

Shailja C Shah, Steven H Itzkowitz. Colorectal Cancer in Inflammatory Bowel Disease: Mechanisms and Management. Gastroenterology. 2022 Mar;162(3):715-730.e3. doi: 10.1053/j.gastro.2021.10.035. Epub 2021 Oct 29.

            - line 54-56: please consider specify that relation between intestinal microorganisms and tumorigenesis is mainly in translational research.

            - line 71-72: consider explain that modulating intestinal microbiota is not a therapeutic aproach in clinical practice (for patients) nowadays.

- Results:

            - line 122: consider explain the meaning of colorectal index in the methods section.

            - line 128: explain as a limitation of the study in the discussion section that only 3 mouse per group were analyzed.

            - line 218-219: consider qualifying (modulate) this phrase because colorectal cancer and ulcerative colitis do not currently have a single known cause. Although, must be explained in the discussion or introduction, nor in the results section.

            - line 289-290: please, consider this sentence more specifically related to translational research.

            - consider explain if there is differences regarding the high dose or low dose of ADe given to the mouse.

MINOR COMMENTS:

- Line 276: there is a double space between 1.0 g/kg and ADe

Author Response

Manuscript ID: Nutrients-2710795.

This study by Lanzhou Li and colleagues describes the effect of the fungus Auricularia delicate in DSS colitis cancer associated.

The strenght of the study are the results obtained (treatment supress inflammatory levels in CAC mice)

Nevertheless, there are some considerations the authors need to review.

MAJOR COMMENTS:

- Introduction:

line 38-39: percentage regarding colorectal cancer in IBD is obtained old studies. Another review about this issue indicate a risk of about 7% at 30 years: Shailja C Shah, Steven H Itzkowitz. Colorectal Cancer in Inflammatory Bowel Disease: Mechanisms and Management. Gastroenterology. 2022 Mar;162(3):715-730.e3. doi: 10.1053/j.gastro.2021.10.035. Epub 2021 Oct 29.

Reply: Thank you for your comment. It has been checked and modified accordingly at line37-38.

line 54-56: please consider specify that relation between intestinal microorganisms and tumorigenesis is mainly in translational research .

Reply: Thank you for your comment. This part has been rewritten accordingly at line 53-65.

line 71-72 : consider explain that modulating intestinal microbiota is not a therapeutic aproach in clinical practice (for patients) nowadays.

Reply: Thank you for your comment. This part has been rewritten accordingly at line 79-83.

- Results:

line 122: consider explain the meaning of colorectal index in the methods section.

Reply: Thank you for your comment. The meaning of colorectal index has been provided at line 125-126.

line 128: explain as a limitation of the study in the discussion section that only 3 mouse per group were analyzed.

Reply: Thank you for your comment. The limitation has been provided accordingly at line 401-403.

line 218-219: consider qualifying (modulate) this phrase because colorectal cancer and ulcerative colitis do not currently have a single known cause. Although, must be explained in the discussion or introduction, nor in the results section.

Reply: Thank you for your comment. It has been checked and modified accordingly.

line 289-290: please, consider this sentence more specifically related to translational research.

Reply: Thank you for your comment. This part has been rewritten accordingly at line 289-302.

consider explain if there is differences regarding the high dose or low dose of ADe given to the mouse.

Reply: Thank you for your comment. The content has been provided in result and discussion.

MINOR COMMENTS:

- Line 276: there is a double space between 1.0 g/kg and ADe

Reply: Thank you for your comment. It has been checked and modified accordingly.

Reviewer 3 Report

Comments and Suggestions for Authors

Main Question Addressed:

The main question addressed by the research is whether the administration of ADe has a protective effect against colorectal cancer in mice with chemically induced CAC and, if so, what mechanisms are involved. The study investigates the impact of ADe on colorectal tumor development, intestinal tissue cells, microbiota composition, serum metabolites, and inflammation mediated by the NF-κB signaling pathway.

Originality and Relevance:

The topic appears to be original and relevant in the field, particularly in the context of colorectal cancer and its association with colitis. The study explores the potential therapeutic effects of ADe, possibly a novel treatment, by addressing both tumor development and the underlying mechanisms involving microbiota and inflammation.

Consistency of Conclusions:

The conclusions drawn appear to be generally consistent with the evidence presented in the study.

 In summary, the research addresses an important question in the field of colorectal cancer and introduces potential therapeutic avenues.

Comments on the Quality of English Language

 English language fine. No issues detected.

Accept in present form.

Author Response

Reply: Thank you very much for your review and comment regarding our manuscript.